# Comparison of Lifestyle of Catholics and Seventh-Day Adventists and the Relationship with Homocysteine as Risk Factor for Cardiovascular Diseases, a Cross-Sectional Study in Polish Males and Females

**DOI:** 10.3390/ijerph18010309

**Published:** 2021-01-04

**Authors:** Anna Majda, Joanna Zalewska-Puchała, Iwona Bodys-Cupak, Alicja Kamińska, Anna Kurowska, Marcin Suder

**Affiliations:** 1Department of Nursing Fundamentals ul, Institute of Nursing and Midwifery, Faculty of Health Sciences, Jagiellonian University Medical College, Michałowskiego 12, 31-126 Krakow, Poland; j.zalewska-puchala@uj.edu.pl (J.Z.-P.); i.bodys-cupak@uj.edu.pl (I.B.-C.); alicja.kaminska@uj.edu.pl (A.K.); anna2.kurowska@uj.edu.pl (A.K.); 2Faculty of Management, AGH University of Science and Technology, Al. Mickiewicza 30, 30-059 Krakow, Poland; msuder@agh.edu.pl

**Keywords:** Catholics, Seventh-day Adventists, lifestyle, homocysteine, cardiovascular disease

## Abstract

Background: A review of epidemiological data demonstrates relationships between defined health behaviours linked with religious affiliation and a reduced risk of developing and dying from Cardiovascular Disease (CVD). The aim of the study was to compare the lifestyle of Seventh-day Adventists (SDA) and Catholics (CA), to determine the relationship between the lifestyle of SDA, CA and the level of serum homocysteine as a risk factor for CVD. Methods: A cross-sectional study was conducted on 252 SDA and CA. The following tools were used: interview questionnaire, anthropometric measurement, the International Physical Activity Questionnaire (IPAQ), the Inventory of Health Behaviours (IHB), the Perceived Stress Scale (PSS-10), laboratory tests (homocysteine level), and the Fagerström Test for Nicotine Dependence (FTND). Results: Selected lifestyle elements, such as smoking cigarettes, drinking alcohol, physical activity, diet, Body Mass Index (BMI), health behaviours on the IHB, psychosocial factors and level of stress for CA were significantly different in comparison to SDA. The religion professed by the respondents was not significantly associated with the increased level of homocysteine as a risk factor for cardiovascular diseases (CVD). The level of homocysteine for CA were significantly different in comparison to SDA. The studied group of CA was significantly influenced by socio-demographic factors: gender, age, education, place of residence, BMI and lifestyle: drinking alcohol and smoking cigarettes, consumption of dark bread, pasta, and groats. For the studied group of SDA, the level of homocysteine was significantly influenced by socio-demographic factors such as gender, age, professional activity, and consumption of legumes. Conclusions: Public health professionals and nurses should develop culturally specific educational interventions.

## 1. Introduction

Religious beliefs can be the source of many health-related habits, which can be looked at in terms of their effect on health [1,2,3,4]. Identification of those components of lifestyle, which prevent diseases in defined human populations, including religious groups, lies within the sphere of interest of public health professionals and nurses. A review of epidemiological data attests to the relationships between defined health behaviours—linked with religious affiliation and religious commitment—and a decreased risk of developing and dying due to, for example, Cardiovascular Disease (CVD). Some studies indicate the existence of a positive relationship between religious intensity and increased physical activity, good nutrition, and cessation of smoking [5,6]. Furthermore, religion can play an important role in the prevention of dependence on alcohol or nicotine [7,8].

Mortality from cardiovascular diseases is associated with atherosclerosis risk factors. They can be divided into modifiable (conventional and new) risk factors and non-modifiable (e.g., age, gender). Modifiable factors are those related to lifestyle (diet, smoking, low physical activity) and biochemical and physiological features (dyslipidaemia, diabetes, hypertension, obesity, metabolic syndrome, thrombotic factors, homocysteine, markers of the inflammatory process).

At the turn of the twentieth and twenty-first centuries, scientists began to take an interest in new risks factors for CVD, such as the serum homocysteine level. Homocysteine is a risk factor for coronary, cerebral and peripheral atherosclerosis. Its concentration in blood depends on gender, age, kidney and liver function, smoking, physical activity and diet. Dietary factors influencing homocysteine levels include the amount of methionine consumed and the supply of folic acid, vitamin B_6_ and B_12._ An elevated level of serum homocysteine contributes to increased platelet aggregation, triggers oxidative stress and proliferation of smooth muscle cells in the arterial wall, and inhibits production of nitric oxide, leading to damage to the vascular endothelium and the development of coronary heart disease [9,10,11,12,13,14,15,16,17,18]. The amount of evidence indicating that psychosocial factors influence cardiovascular risk is also increasing [19,20,21,22].

Due to the lack of research determining the risk factors of cardiovascular diseases among religious groups, the authors of the article decided to join the group of researchers dealing with the old (lifestyle, dyslipidemias, blood pressure, blood glucose levels, obesity) and new (homocysteine levels) cardiovascular risk factors among adherents of different religions. This manuscript is one of a series of papers presenting the results from studies that were conducted in a group of Catholics (CA) and Seventh-day Adventists (SDA) living in southern Poland [23,24,25,26,27,28]. In this text we present only comparing the lifestyles of Seventh-day Adventists and Catholics and determining the relationship between lifestyle and serum homocysteine levels.

## 2. Materials and Methods

### 2.1. The Aim of the Study

The main aim of the study was to compare the lifestyle of Seventh-day Adventists (SDA) and Catholics (CA), to determine the relationship between the lifestyle of SDA, CA and the level of serum homocysteine as a risk factor for Cardiovascular Disease (CVD).

This study includes as specific objectives: (1) a comparison of lifestyle of ADS and CA (the level of health behaviour on the IHB scale; the level of perceived stress on the PSS-10 scale; eating habits, physical activity, addictions; psychosocial factors such as work stress, social isolation, hostility, type D personality, anxiety); (2) determining the relationship between SDA and CA lifestyle and the level of homocysteine in the blood serum and its comparison.

### 2.2. Organisation and Course of the Study

The cross-sectional research was conducted in the years 2014–2015 among 252 people [24,25]. The selection of the studied group was targeted, and the sample size dictated by the project’s financial resources. The SDA and CA were reached through notices/advertisements informing about the conducted study, and the schedule was established by pastors or priests. Subjects were recruited from Polish SDA and CA living in southern Poland. The respondents were assured of anonymity, informed of the study’s assumptions and its course, voluntary participation, and the opportunity to withdraw from participation at any stage of the study, receiving “Information for the respondents” in writing. They gave their informed consent to participate in the study by completing and signing the “Respondents Consent Form” and “Personal Data Processing Form” before proceeding with the project. Persons who practiced their religion were identified by making a statement/declaration.

The criteria for inclusion in the study were over 18 years of age and practising the SDA or CA religion. Exclusion criteria were: not practising the given religion, pregnancy, breastfeeding, autoimmune diseases, cancer, and operations within the last 3 weeks.

### 2.3. The Study Group

The cross-sectional study was conducted in a group of 252 people took part in the study—118 Seventh-day Adventists (SDA) and 134 Catholics (CA) living in southern Poland. Comparing socio-demographic variables in the two groups, it can be observed that both SDA and the CA were most often inhabitants of towns (SDA 66.9% vs CA 78.4%) and were mostly women (SDA 63.6% vs CA 64.9%). Their average age was similar (SDA 53 years—age 24–94 years vs CA 47.7 years—age 20–96 years). They were most frequently professionally active (SDA 69.5% versus CA 79.9%), white collar workers (SDA 52.4% vs CA 67.2%), and their main source of income was their employment (SDA 66.1% vs CA 70.2%). They differed in terms of education: SDA usually had secondary education, whilst CA—higher education (Table 1). SDA more frequently suffered from chronic diseases (SDA 64.4% vs CA 47.8%). Subjects in both groups declared that they regularly took medical drugs (SDA 93.5% vs CA 96.5%). None of the subjects took folate [24,25].

### 2.4. Methods, Techniques and Research Tools

The following research tools were used as part of the diagnostic survey method: a questionnaire survey designed by us; the International Physical Activity Questionnaire (IPAQ) by M. Sjöström, B. Ainsworth, A. Bauman, F. Bull, C. Craig, and J. Sallis [29]; the Inventory of Health Behaviours (IHB) by Z. Juczyński [30]; the Perceived Stress Scale (PSS 10) by S. Cohen, T. Kamarck, and R. Mermelstein, adapted by Z. Juczyński, and N. Ogińska-Bulik [31]; the Fagerström Test for Nicotine Dependence (FTND) [32].

The interview questionnaire contained 57 questions. It was divided into three parts, Part A related to addictions and eating habits; part B encompassed psychosocial factors influencing cardiovascular disease; part C contained socio-demographic data relating to the respondent. Part A contained questions related to the consumption of alcohol (type: wine, beer, vodka, the number of standard serving sizes consumed and the frequency: I do not drink alcohol, I drink occasionally/very rarely—not more than once a month, rarely —several times a month or once a week, often/frequently—several times a week, daily smoking cigarettes (how many cigarettes daily, for how many years).

Anthropometric measurements encompassed body mass and height. Height was measured to the nearest 0.5 cm using a stadiometer. Body weight was measured to the nearest 0.1 kg, using certified portable electronic medical scales. For each person, a Body Mass Index (BMI) was calculated. Overweight was defined as BMI 25–29.9 kg/m^2^, obesity as BMI ≥ 30 kg/m^2^. The above standards are consistent with European recommendations concerning prevention of cardiovascular disease in clinical practice, 2012 (European guidelines on cardiovascular disease prevention in clinical practice, version 2012).

The International Physical Activity Questionnaire [29] described physical activity in units of energy expenditure MET-minutes/week (MET—Metabolic Equivalent of Task is a metabolic unit measured in relation to the consumption of oxygen at rest: 3.5 mL O_2_/kg body mass/min). Determining the level of physical activity of a person involves calculating the total energy expenditure based on multiplying the frequency and duration of exercise by the appropriate intensity expressed in MET units. The short version of the IPAQ, containing seven questions about overall physical activity, was used in the study. The level of weekly physical activity was divided into three categories: insufficient physical activity (<600 MET-min/week); sufficient physical activity (600–1500 MET-min/week); high physical activity (>1500 MET-min/week).

Inventory of Health Behaviours [30] contains 24 statements describing various behaviours linked with health. The behaviours were divided into four categories: good dietary habits, preventive behaviours, health practices and a positive mental attitude. Respondents choose answers to each question in the form a 5-item Likert item. The general result IHB is converted into standardized units and interpreted according to a sten scale, where sten scores within the range 1–4 are considered low, 5–6: average, and 7–10: attesting to a high intensity of health behaviours.

Questions on the PSS-10 Perceived Stress Scale [31] relate to assessment of the intensity of stress linked with the respondent’s own life situation. Respondents answered each question on a five-item Likert item. Overall results were converted into standardized units and interpreted in accordance with the scale key, Sten scores ranging from 1 to 4—a low result; from 7 to 10—high; and in the range 5–6—average.

The Fagerström Test for Nicotine Dependence [32] consists of six questions, on which you can get up to a maximum of 10 points. A result below 7 points—behavioural (psychological) dependence; a result above 7 points—pharmacological (biological) dependence.

Blood that was to be tested for homocysteine was collected and transported in accordance with recommendations, all subjects were fasting. The normal range for homocysteine is 0–12 µmol/L.

In this article, the analysis of the influence of selected factors on the level of homocysteine in the blood serum and the differentiation of this influence depending on the religion was carried out in a two-stage form. In the first part of the analysis, it was verified whether there was a difference in the percentage of people with increased levels of homocysteine in the blood serum between Catholics and Adventists, taking into account factors/predictors such as age, gender, education, place of residence, work, health behaviour (smoking cigarettes, drinking alcohol, Statistical analysis was based not only on the presentation of data in tabular form but also on the use of a statistical test to verify whether the indicated differentiation is statistically significant. For this purpose, the Chi-square test of independence—Chi^2^ (Pearson’s test) was used. The level of significance was *p* ≤ 0.05 [33]. In the second part of the analysis of the influence of selected factors on the level of homocysteine in the blood serum, a logistic regression model was used. The above-mentioned model is used in a situation where the dependent variable is measured on a nominal scale and has two values. The nature of the dependent variables can be qualitative or quantitative. This analysis can be considered as an alternative to classical regression models in which the dependent variable is quantitative [34]. In the presented considerations, it was possible to use this model due to the dichotomous nature of the dependent variable, that is, the level of homocysteine (0—normal or 1—above the norm), adopted in the study. A significance level of 5% was adopted in the statistical analysis carried out in the study for the verification of hypotheses and for testing the significance of relevant coefficients. Statistical analyses were performed using the Statgraphics 18 program. The use of the logistic regression model allows determining both the strength and direction of the relationship between a qualitative (class type) or quantitative (discrete or continuous type) factor and a dichotomous explanatory variable. Apart from the dichotomous independent variable, a necessary condition for the use of logistic regression is a sufficiently large sample, which should be greater than 10 × (k + 1), where k—is the number of independent variables [35]. In order to infer the influence of independent variables on the dependent variable, the odds ratio was used. It determines the change in the odds of the singled-out value of 1 of the dependent variable when the independent variable increases by 1 unit. It is assumed that the remaining independent variables in the model remain on the same level, while the tested independent variable increases by one unit.

### 2.5. Ethical Considerations

Ethical approval was obtained from the Academic Bioethics Committee (KBET/79/B/2014). Participants were provided a written participant document of the study purpose, benefits, potential risk and voluntary withdrawal.

## 3. Results

### 3.1. Homocysteine Levels

An elevated level of homocysteine was diagnosed in over half of the studied (SDA 55.9%; CA 60.5%). Amongst men considerably more frequently had an elevated result (SDA 72.1%; C 78.7%) and respondents aged > 60 years (SDA 72.2%; CA 83.3%) (Table 2). The average value of homocysteine concentration for the whole SDA group was 13.7 µmol/l and for the whole CA group was 16 µmol/l. Comparing the two groups in terms of the mean reference values of homocysteine given above, it is noticeable that it is significantly higher in Catholics (*p* = 0.0430) (Majda et al., 2015; 2017a; 2017b).

### 3.2. Eating Habits and BMI

The vast majority of respondents in the two compared groups consumed 3–4 meals in the course of a day (C 81.3%; SDA 82.2%). Significant differences (*p* < 0.0000) were observed in terms of consumption of main meals. Breakfast was consumed by 10.2% SDA and 88.8% CA; Packed lunch/elevenses: 34.7% SDA and 55.2% CA; lunch: 95.8% SDA and 89.5% CA; tea: 19.5% SDA and 23.9% CA, and supper: 68.6% SDA and 79.8% CA. The biggest meal of the day for both studied groups was lunch (SDA 56%; CA 62.7%), followed by breakfast (SDA 41.5%; CA 23.9%), and for 2.5% SDA and 13.4% CA—supper; the differences were statistically significant (*p* = 0.0030). The last meal was consumed: before 18:00 by 31.4% SDA and 23.1% CA; between 18:00 and 20:00 (SDA 50.8%; CA 42.6%); between 20:00 and 22:00 (SDA 17.8%; CA 31.3%), and after 22:00 only by 3.0% CA (statistically significant differences, *p* = 0.0399). Unfortunately, respondents from both groups admitted that they snack between main meals (SDA 64.4%; CA 73.9%).

Significant differences (*p* < 0.0000) were observed in terms of how meals were prepared. The majority of both the studied Adventists and Catholics preferred preparing meals by boiling and baking, although boiling was more frequently preferred by SDA (83.9% vs. 69.4%), and baking by CA (62.7% vs. 37.3%).

The Seventh-day Adventists diet differed significantly from that of the Catholics in the area of consumption of bread, rice, pasta, groats, meat, fats, legumes and offal (organ meats). Analysis of preferences of selected food products showed that more studied Seventh-day Adventists than Catholics consumed brown bread, skimmed/low-fat dairy products, brown rice, pasta, multi-grain groats, multi-grain and wheat flakes, white fish, vegetable fats. However, more Catholics than Seventh-day Adventists consumed: wheat bread, full fat/whole dairy products, both full fat/whole milk and skimmed/low fat milk, white rice, pasta, wheat groats, oily fish, meat (poultry, beef, and pork), and animal fats. Furthermore, a larger percentage of studied SDA than CA consumed legumes, eggs, and vegetables, whilst more CA consumed offal (Table 3).

The concentration of homocysteine in serum among SDA who declared the consumption of legumes and brown bread was significantly more frequently within the normal range (*p* = 0.0062; *p* = 0.0282). The concentration of homocysteine in serum among CA who declared the consumption of full fat of full fat (whole) milk, multi-grain flakes and pasta, and wheat groats was significantly more frequently within the normal range (*p* = 0.0498; *p* = 0.043; *p* = 0.0177).

Comparing BMI in the two groups, a somewhat higher percentage of SDA was within the normal range. Among CA, almost twice as many individuals were obese, whilst the percentage of overweight subjects in both groups was the same. The differences between the results obtained for BMI in the two studied groups were not significant (Table 4).

### 3.3. Addictions

All SDA respondents declared that they did not drink alcohol or smoke cigarettes. Amongst the studied CA, 24.6% did not consume alcohol, 30.7%—very rare consumption, rare—56.4%, frequent—11.9%, and every day—1.0% (Table 5). The subjects most frequently drank wine (59.7%), then beer (49.3%) and vodka (37.3%); they most frequently declared that the amount they consumed was a single standard unit of alcohol (at a time), 17.2% of subjects smoked cigarettes (M 48%; W 52%). Within the group of smokers, the largest subgroup, 17.4%, had smoked for 20 years; somewhat over half of (smoking) subjects (52.2%) smoked 10 or less cigarettes a day, and just under half (43.5%): 11-20 cigarettes. Almost all (smoking) subjects gained less than 7 points on the Fagerström Test for Nicotine Dependence, so their degree of addiction was of a psychological nature. Non-smokers borderline more frequently had a normal level of homocysteine (*p* = 0.0515).

### 3.4. Physical Activity

Somewhat over half of the studied SDA and almost half of the studied CA (59.3% vs 46.3%) stated that they engaged in a high level of physical activity. Moderate effort, which is considered sufficient activity, was declared by 30.5% of studied SDA and slightly fewer CA (28.3%), while for 10.2% of the studied Seventh-day Adventists and 25.4% of the studied Catholics, the level of physical activity was insufficient (Table 6). Statistical analysis showed that insufficient physical activity was significantly (*p* = 0.0065) more common among CA than SDA, and also did not significantly influence the obtained results for serum homocysteine levels (SDA—*p* = 0.9114; CA—*p* = 0.3716).

### 3.5. Health Behaviors

Sten analysis of the results for the intensity of health behaviours of subjects on the Inventory of Health Behaviours allows us to ascertain that the intensity of health behaviours of the Seventh-day Adventists was, for most of them, high (54.2%) and average (36.4%), and low for only 9.3% of them. On the other hand, among CA, the intensity of health behaviours was for most of them average (50.8%) and low (26.1%), but high for 23.1%. Statistical analysis showed that the intensity of positive health behaviours among SDA on the Inventory of Health Behaviours was significantly higher than among CA (*p* = 0.0087), and was not associated with the level of homocysteine in the serum (SDA—*p* = 0.1841; CA—*p* = 0.6654). Analysis of results in particular areas (subscales) showed that the most highly rated categories of health behaviours by SDA were good eating habits and a positive mental attitude; preventive behaviours and health practices were rated somewhat lower. In contrast to SDA, CA indicated health practices most rarely. The most highly rated category of health behaviours by them was, however, a positive mental attitude (Table 7).

### 3.6. Perceived Stress

Analysis of results on the PSS-10 Scale showed that a high level of stress was more common amongst studied Catholics (CA 31.9% vs SDA 21.2%). Both surveyed Catholics and Seventh-day Adventists were most frequently characterized by an average level of intensity of stress (CA 34.8%; SDA 42.4%). A low level of stress was observed in 33.3% of CA and 36.4% of SDA. The results obtained on the PSS-10 in the studied group of CA were in the range of 2 to 10 sten, and amongst SDA: from 2 to 9 sten. Table 8 presents a detailed breakdown of replies by studied SDA and CA on the PSS-10 Scale. Statistical analysis showed an insignificant difference between the level of stress experienced by SDA and CA (*p* = 0.1512), and also that the stress level did not have a significant relationship with values for the level of homocysteine in serum (SDA—*p* = 0.4643; CA—*p* = 0.5642).

### 3.7. Psychosocial Factors

Psychosocial factors constituted a risk factor for cardiovascular disease for one tenth of SDA (10.2%) and for one quarter of CA (24.6%). In first place among SDA was anxiety (27.9%), in second place—hostility (23.7%), in third—a type D personality and a tendency to social isolation (17.8% and 17.4% respectively), and in fourth—stress at work (11%). On the other hand, most surveyed CA experienced hostility (41%), then a high level of anxiety (36.2%), had a type D personality (28.7%), a tendency to social isolation (13.8%) and stress at work (10.4%) (Table 9). Statistical analysis showed that psychosocial factors constituted a risk factor for cardiovascular diseases significantly more frequently (*p* = 0.0024) for CA than for SDA, but they did not have a significant association with homocysteine level results either in the serum of SDA (*p* = 0.6623) or CA *(p =* 0.6661).

Due to the fact that one of the main goals of the study was to verify the influence of religion on the level of homocysteine, a univariate logit model was created, in which only the type of religion was assumed as an independent variable. Table 10 shows the results of this analysis. The Logistic Regression Score (LRS) is an estimate of the relationship between the religion variable and the probability of elevated homocysteine levels. The presented results clearly show that there is no significant relationship between the level of homocysteine and the type of religion (SDA or CA). For the entire model, the value of *p* is greater than 0.05. Additionally, it can be seen that the obtained value of 0.2513802 of the coefficient in the model with the religion variable is not statistically significant.

Therefore, logistic regression analysis was performed separately for SDA and CA data. In this way, it was possible to diagnose which variables significantly influence the level of homocysteine for individual religious groups. Table 11 shows the results of the logistic regression analysis for CA. On their basis, it can be concluded that with the assumed form of dependence, the influence of some of the variables turned out to be significant. The variables for which the significance of the coefficient for the model was obtained are consumption of alcohol, BMI, sex, age, place of residence, and education. When it comes to smoking cigarettes, the result was inconclusive. Although the test probability *p* equal to approximately 0.052 is greater than the adopted significance level, the value of the statistics for the entire model indicates its correctness. Therefore, it can be assumed that smoking is also an important factor influencing the level of homocysteine. When interpreting the obtained results, for example, for the impact of BMI on the level of homocysteine, we can conclude that as body weight increases the risk of elevated homocysteine levels increases. This is evidenced by the positive value of the estimated coefficient, that is, 0.517. At the same time, based on the odds ratio (OR), we can conclude that with an increase in the BMI value, i.e., for example, with the transition from normal body weight to overweight, the probability of having an increased level of homocysteine increases by about 1.7 times (OR = 1.677). An unexpected result was obtained for the consumption of alcohol. The negative value of the coefficient for the alcohol variable indicates that more frequent alcohol consumption lowers the risk of increased homocysteine levels. To sum up, with age, weight gain and for people living in the city, as well as for people with lower education and men, the probability of having an increased level of homocysteine is relatively the highest.

Table 12 shows the results of univariate logistic regression analysis for CA in the scope of selected food products. When it comes to Catholics—dark bread, groats, and pasta were significant variables in the model used. The obtained results show that not consuming these products increases the risk of belonging to a group of people with an increased level of homocysteine. Based on the OR value for these products, it can be concluded that people who do not eat them are more than twice as likely to have an increased homocysteine value than people who eat these products.

Table 13 shows the results of the logistic regression analysis for SDA. In the case of Adventists, the important variables in the applied univariate logistic regression model are gender, age, and professional work. The direction of the obtained dependence is different than in the case of Catholics. For male Adventists, the risk of developing increased levels of homocysteine was more than three times higher than for women. Additionally in the case of SDA, unexpected results were obtained for the age and professional activity variables. The risk of increased homocysteine value decreased for the elderly and the unemployed.

Table 14 shows the results of the logistic regression analysis for the SDA for selected food products. In the case of ADS, only legumes were significant variables in the applied logistic regression model. People who did not eat legumes were seven times more likely to find elevated levels of homocysteine.

## 4. Discussion

Many researchers throughout the world are interested in which elements of lifestyle based on religious commitment are significant for health. The question arises as to how lifestyle, including dietary habits, physical activity, exposure to stress, and psychosocial factors influence the assessment of the risk of cardiovascular diseases among followers of different religions, including members of the religious denomination that has the greatest longevity—SDA [36,37,38,39,40,41,42]. The role of religion (religiosity) in coping with various health and sickness situations has also begun to be perceived in recent years in the Polish literature [43], although there are no studies on this subject in relation to particular religious groups. We decided to compare the results of our own research with those of nationwide (Polish) studies such as NATPOL PLUS from 2002 and WOBASZ from 2003–2005 [17,44,45].

In this study we found that the SDA diet differed significantly from the CA diet in terms of consumption of bread, rice, pasta, groats, meat, fats, legumes and offal (organ meats). The concentration of homocysteine among SDA who declared the consumption of legumes and brown bread was significantly more frequently within the normal range. The concentration of homocysteine among CA who declared the consumption of full fat milk, multi-grain cereals, and wheat pasta and wheat groats were significantly more frequently within the normal range. The applied logistic regression model based on the odds ratio (OR) values showed that for SDA who did not eat a lot of legumes, the risk of increased homocysteine levels significantly. For CA, who did not eat a lot of dark bread, groats, pasta, it was possible to find a higher probability of increased level of homocysteine.

In this study, as in other epidemiological studies [4], SDA were characterised by desirable health behaviours and regular religious practices. They did not drink alcohol or smoke cigarettes. Their desirable health behaviours included a diet that was similar to a vegetarian one. Unfortunately, more than half of subjects SDA admitted to snacking between main meals, and only one tenth to eating breakfast. Among Catholics surveyed were drinkers and cigarette smokers. Catholics’ diet abounded in products of animal origin (including meat and fat). Almost 3/4 of subjects Catholics admitted to snacking between main meals. In the NATPOL PLUS study also the average diet of adult Polish citizens contains a high amount of total fat and saturated fatty acids, as well as amounts of unsaturated fatty acids and dietary cholesterol that are close to the recommended intakes [45].

Over half of both the studied Catholics and Seventh-day Adventists had elevated levels of homocysteine in the serum. The one-way logistic regression analysis showed that religion professed by the respondents was not significantly associated with the increased level of homocysteine as a risk factor for cardiovascular diseases (CVD). In the light of reports by researchers, the level of homocysteine in blood serum is influenced by such factors as: diet (meat, eggs, milk, products rich in vitamin B_12_, B_6_, and folic acid), various diseases (including kidney and liver failure, diabetes, hypothyroidism, and lymphoblastic leukaemia), alcohol, smoking, coffee drinking, taking medicines (including anticonvulsants—phenytoin, cytostatic drugs—methotrexate, antidiabetic medication—metformin, and oral hormonal contraceptives) [11]. It could be expected that SDA respondents who did not smoke, did not drink alcohol, did not take any of the above-mentioned medicines, and did not have the above mentioned diseases (with the exception of diabetes in certain cases) should have a low level of homocysteine. The fact that it was elevated for 55.9% SDA respondents can be explained by the following: a factor that could have had an influence on the increase was the consumption of a significant amount of eggs and milk—supplying the body with large amounts of methionine; or by the fact that their diet, which was close to being vegetarian, generally contained smaller amounts of folates and B vitamins, which could only confirm the significant relationship between the high level of homocysteine in some of the studied SDA and their low consumption of legumes, which are a rich source of folate, and brown bread, which contains a lot of B vitamins. A systematic review of the research conducted by Afshin et al. [46] showed the relationship between consumption of legumes and the lower risk of coronary heart disease. The fact that consumption of full fat (whole) milk, multi-grain flakes and wheat pasta and groats had an influence on the level of homocysteine in Catholics can be explained by the fact that the supply of folate in the diet, a source of which is (apart from yeast, green vegetables, grain legumes, bran, liver, poultry and eggs), milk products, whole grain wheat or rye bread [11] —which were consumed (especially the latter) by the studied Catholics—plays a role in the prevention of elevated homocysteine.

There are no reports in the scientific literature about the level of physical activity and stress experienced among the SDA and CA and their association with the concentration of homocysteine in the blood serum does not allow for comparison of results. In the logistic regression model, smoking cigarettes by CA turned out to be a significant factor influencing levels of homocysteine, as expected and scientifically reported. An unexpected result was obtained for drinking alcohol. The negative value of the logistic regression coefficient for the alcohol variable indicates that more frequent consumption of alcohol by CA reduces the risk of increased levels of homocysteine. On the basis of the odds ratio (OR) index, it was also found that as the BMI in CA increases, the probability of having an elevated level of homocysteine increases.

Meta-analyses have shown that, amongst other things, a low level of education, social isolation and little social support, stress at work and in family life, symptoms of anxiety and hostility, and a type D personality (characterized by frequent nervousness, irritation, and dejection, and by avoiding the sharing of emotions) increase the risk of CVD [19]. In the studied group of SDA, these factors related to one tenth of respondents, whereas among Catholics—to one quarter. Thus, these two groups differed significantly in terms of these factors, but none of these factors had a significant relationship with the level of homocysteine. The results of the WOBASZ study showed that the frequency of occurrence of negative psychosocial factors differs significantly between the genders to the disadvantage of women [44].

The results of this study are preliminary. Further research needs to be conducted—on the influence of lifestyle on homocysteine levels among SDA and CA—on a larger group in order to verify the results of the present study. The research needs to be continued as the results are inconclusive and contradict the previous reports on the relationship between the homocysteine levels, and age, professional activity, drinking alcohol.

Although the research provided important information on the subject of lifestyle and homocysteine levels among SDA and CA, several limitations of the study need to be taken into account when analysing the results. These include the small size of the studied group, the lack of genetic testing, due to the limited financial resources of the project and the high cost of biochemical and genetic tests. Also, the number of participants was relatively small in compared to the large heterogeneity of their age (from 18 to 53 years of age), and people with cardiovascular, cerebrovascular and lung diseases, both CA and SDA, were not excluded from the study.

The results of CA and SDA taking medications were included in the analysis. The subjects in both groups declared their regular intake, most often they were antihypertensive and hypoglycaemic drugs that could affect the final test results. CA and SDA differed in education. SDA most often had secondary education and CA had higher education. According to European guidelines for the prevention of cardiovascular disease in clinical practice, people with low levels of education have higher CVD mortality [19]. Despite the higher education of Catholics, other psychosocial variables, such as anxiety, hostility, and type D personality, have proved to be a risk factor for cardiovascular disease more often in CA than SDA. Thanks to the use of the logist regression model in the presented study, it was possible to prove that for older CA, living in the city, with lower education, males, the probability of having increased levels of homocysteine as expected was significantly higher. On the other hand, for SDA, the risk of having increased homocysteine value was lower for the elderly and the unemployed, which was unexpected.

Both public health professionals and nurses should therefore develop educational interventions related to culture and religion in order to promote healthy lifestyle principles in the prevention of CVD, especially among Catholics.

## 5. Conclusions

The religion professed by the respondents was not significantly associated with the increased level of homocysteine as a risk factor for cardiovascular diseases (CVD).

The level of homocysteine in the studied group of Catholics was significantly influenced by socio-demographic factors: gender, age, education, place of residence, BMI and lifestyle: alcohol consumption and smoking cigarettes, consumption of dark bread, pasta and groats.

The level of homocysteine in the studied group of SDA was significantly influenced by socio-demographic factors: gender, age, professional acivity and consumption of legumes.

## Figures and Tables

**Table 1 ijerph-18-00309-t001:** Sociodemographic data.

Variables		CA N—134		SDA N—118	
		N—134	%	N—118	%
Gender	Female	87.0	64.9	75.0	63.6
	Male	47.0	35.1	43.0	36.4
Place of residence	Village	29.0	21.6	39.0	33.1
	City	105.0	78.4	79.0	66.9
Education	Primary/Elementary	7.0	5.2	6.0	5.1
	Vocational	18.0	13.4	32.0	27.1
	Secondary	33.0	24.6	41.0	34.7
	Higher	79.0	56.8	39.0	33.1
Professional work/Professional activity	Physical work	17.0	12.7	39.0	33.1
	Intellectual work	90.0	67.2	43.0	36.4
	Not working/Unemployed	27.0	20.1	36,0	30.5
Source of income	Professionally active	94.0	70.2	75.0	63.6
	Disablement pension	6.0	4.5	7.0	5.9
	Retirement pension	30.0	22.4	32.0	27.1
	Benefits	4.0	2.9	4.0	23.4
		Range	M (SD)	Range	M (SD)
Age		20–96	47.7 (16.039)	24–94	53 (15.711)

Note. Key: N—number of subjects; M—mean; SD—standard deviation.

**Table 2 ijerph-18-00309-t002:** Homocysteine results in the study group of SDA and CA: overall, according to age and to gender.

**Results**	**Gender**				**Age**						**Overall (%)**	
	**Female (%)**		**Male (%)**		**SDA**	**CA**	**SDA**	**CA**	**SDA**	**CA**		
	SDA	CA	SDA	CA	under 39 years of age	under 39 years of age	40–60 years old	40–60 years od	60+	60+	SDA	CA
**Homocysteina**												
Above normal	46.7	50.6	72.1	78.7	56.5	45.8	45.8	58.0	72.2	83.3	55.9	60.5
*p*	χ^2^ = 0.25; *p* = 0.62		χ^2^ = 0.54; *p* = 0.64		χ^2^ = 0.71; *p* = 0.39		χ^2^ = 1.62; *p* = 0.20		χ^2^ = 1.29; *p* = 0.26		χ^2^ = 0.53; *p* = 0.47	

Note. *p*—statistical value.

**Table 3 ijerph-18-00309-t003:** Preferences of studied CA and SDA concerning consumption of selected food products.

Product	% of Persons Consuming Product			
		CA	SDA	*p*
Bread	Wheat	50.7	29.6	0.0044 *
	Brown	66.4	85.6	
Dairy products	Full fat/whole	51.5	43.2	0.3487
	Skimmed/low fat	48.5	54.2	
Milk	Full fat/whole	42.2	19.5	0.3803
	Skimmed/low fat	45.5	27.9	
Rice	Brown	21.6	50.0	0.0036 *
	White	79.1	63.5	
Eggs		76.9	83.0	0.5848
Pasta, groats	Multi-grain	55,2	78,8	0.0036 *
	Wheat	51.5	33.1	
Flakes	Multi-grain	51.5	68.6	0.8572
	Wheat	14.9	17.8	
Fish	Oily	55.2	35.6	0.1196
	White	57.5	58.5	
Meat	Poultry	88.8	66.1	<0.0000 *
	Beef	62.7	27.1	
	Pork	68.7	0.8	
Fat/oil	Vegetable	76.1	90.7	0.0003 *
	Animal	45.5	18.4	
Vegetables		88.8	96.4	0.6021
Legumes		70.1	86.4	0.0063 *
Offal		41.8	8.5	<0.0000 *

Note. * Chi-squared test; *p*—statistical value; * *p* ≤ 0.05.

**Table 4 ijerph-18-00309-t004:** Results of BMI in the study group of SDA and CA: overall, according to gender.

		Underweight	Normal	Overweight	Obesity	*p*
		%	%	%	%	
CA	Female N—87	3.45	47.13	31.03	18.39	
	Male N—47	-	31.90	40.43	27.67	χ^2^ = 8.75; *p* = 0.12
	Overall N—134	2.24	41.79	34.33	21.64	
SDA	Female N—75	1.33	56.01	32.00	10.66	
	Male N—43	-	44.18	39.54	16.28	
	Overall N—118	0.85	51.65	34.75	12.75	

Note. *p*—statistical value.

**Table 5 ijerph-18-00309-t005:** Addiction among SDA and CA.

Frequency of Alcohol Consumption	SDA	CA
	%	%
I do not drink alcohol	0	24.6
Sporadically/very rare—not more than once a month	0	30.7
Rarely—several times a month or once a week	0	56.4
Often/frequent—several times a week	0	11.9
Daily	0	1.0
Smoking cigarettes:	0	17.2
10 or less cigarettes a day	0	52.2
11–20 cigarettes a day	0	43.5

**Table 6 ijerph-18-00309-t006:** Physical activity among SDA and CA.

Physical Activity	SDA	CA	*p*
	%	%	
High level of physical activity (>1500 MET-minutes/week)	59.3	46.3	0.0065 *
Sufficient level of physical activity (600–1500 MET-minutes/week	30.5	28.3	
Insufficient level of physical activity (<600 MET-minutes/week)	10.2	25.4	

Note. * *p*—statistical value; * *p* ≤ 0.05.

**Table 7 ijerph-18-00309-t007:** Mean results for SDA and CA on particular subscales of the Inventory of Health Behaviours (IHB).

Subscales of IHB		Catholics (CA)			Seventh-Day Adventists (SDA)		
		Female (*n* = 87)	Male (*n* = 47)	Overall (*n* = 134)	Female (*n* = 75)	Male (*n* = 43)	Overall (*n* = 118)
Good eating habits	M	3.54	2.96	3.34	4.24	3.93	4.91
	SD	0.79	0.81	0.84	0.51	0.73	0.62
Preventive behaviours	M	3.61	3.24	3.48	3.81	3.28	3.61
	SD	0.66	3.44	0.67	0.61	0.76	0.71
Positive mental attitude	M	3.66	3.44	3.58	3.94	3.93	3.86
	SD	0.68	0.63	0.67	0.51	0.6	0.55
Health practices	M	3.4	3.12	3.3	3.28	3.73	3.21
	SD	0.77	0.62	0.73	0.66	0.72	0.69

Note. M—mean, SD—standard deviation, *n*—number of subjects.

**Table 8 ijerph-18-00309-t008:** Detailed breakdown of answers by studied SDA and CA on the PSS-10 Scale.

Questions on the PSS-10 Scale	Never (%)		Almost Never (%)		Sometimes (%)		Fairly Often (%)		Very Often (%)	
	CA	SDA	CA	SDA	CA	SDA	CA	SDA	CA	SDA
In the last month, how often have you been upset because of something that happened unexpectedly?	8.89	9.32	12.59	23.72	42.96	45.76	25.93	18.64	9.63	2.54
In the last month, how often have you felt that you were unable to control the important things in your life?	20.75	33.05	33.33	23.72	29.63	31.35	11.85	10.16	4.44	1.69
In the last month, how often have you felt nervous and “stressed”?	2.96	8.47	12.59	18.64	43.70	43.22	26.67	27.11	14.08	2.54
In the last month, how often have you felt confident about your ability to handle your personal problems?	7.40	12.71	6.66	7.62	23.70	19.49	36.30	40.67	25.94	19.49
In the last month, how often have you felt that things were going your way?	2.22	4.23	5.19	9.32	35.56	27.96	43.70	40.67	13.33	17.79
In the last month, how often have you found that you could not cope with all the things that you had to do?	16.30	17.79	31.11	25.42	30.37	35.59	18.52	16.10	3.70	7.62
In the last month, how often have you been able to control irritations in your life?	1.49	5.93	2.96	4.23	31.11	27.11	42.22	44.91	22.22	17.79
In the last month, how often have you felt that you were on top of things?	5.19	3.38	7.40	7.62	36.30	37.28	40.00	37.28	11.11	14.40
In the last month, how often have you been angered because of things that were outside of your control?	9.63	21.18	22.22	29.66	41.49	38.13	20.00	9.32	6.66	1.69
In the last month, how often have you felt difficulties were piling up so high that you could not overcome them?	17.04	26.27	33.33	31.35	31.85	33.05	14.08	9.32	3.70	0.00

**Table 9 ijerph-18-00309-t009:** Psychosocial risk factors for CVD among studied SDA and CA.

Psychosocial Variables		Seventh-Day Adventists (SDA)		Catholics (CA)	
		YES	NO	YES	NO
		(%)	(%)	(%)	(%)
Stress at work	Lack of control over way of fulfilling duties at work	11.01	88.98	10.40	89.60
Social isolation	Living alone	15.25	84.74	11.90	88.10
	Feeling a lack of someone close in whom you can confide	19.49	80.51	15.70	84.30
Type D personality	Experiencing dejection, depression or feeling of hopelessness	12.72	87.28	26.10	73.90
	Loss of interest in life and its pleasures	6.78	93.22	19.40	80.60
	Frequently feeling irritable, dejected	18.64	81.36	35.10	64.90
	Avoiding sharing thoughts and feelings with other people	33.05	66.95	34.30	65.70
Anxiety	Frequently feeling nervous/agitated, uneasy	26.28	73.72	36.60	63.40
	Experiencing difficulty in controlling worries	29.66	70.34	35.80	64.20
Hostility	Frequently getting angry over minor issues	23.73	76.27	41.80	58.20
	Frequent nervousness/agitation due to the habits of others	23.73	76.27	40.30	59.70

**Table 10 ijerph-18-00309-t010:** Logistic Regression Score (LRS)—assessment of the relationship between the religion variable and the probability of increased homocysteine levels.

Parameter	Estimate	Standard Error	Chi-Square (Walda)	*p*	Odds Ratio (OR)	Statistics for the Model
Constant	0.2040953	0.185036	1.216615	0.2700342	1.226415	Chi^2^ = 0.96336; *p* = 0.32635
Religion—X_1_	0.2513802	0.25623	0.9625035	0.3265643	1.285799	

Note. *p*—statistical value. The following variables and their categories were adopted in the study: Y—dependent variable of a dichotomous nature (0—normal homocysteine level, 1—increased homocysteine level). The first dependent variable was the variable X1—defining the religion of the participant of the study, in this case, the following coding was adopted: 0—SDA, 1—CA).

**Table 11 ijerph-18-00309-t011:** Logistic regression score (LRS)—assessment of the relationship between variables considered individually and the probability of an increased level of homocysteine for the CA group.

Parameter	Estimate	Standard Error	Chi-Square (Walda)	*p*	Odds Ratio (OR)	Statistics for the Model
Constant	0.3031863	0.1903103	2.538015	0.1111443	1.354167	Chi^2^(1) = 4.4554 *p* = 0.03480
Smoking cigarettes (X_2_)	1.143733	0.5874138	3.79106	0.05153573	3.138462	
Constant	1.260634	0.362727	12.07867	0.000511	3.527656	Chi^2^(1) = 7.7458 *p* = 0.00539 *
Drinking alcohol (X_3_)	−0.912754	0.34413	7.034986	0.007997	0.401417	
Constant	0.070285	0.246257	0.081461	0.77533	1.072814	Chi^2^(1) = 4.9516 *p* = 0.0260 *
BMI (X_4_)	0.517393	0.238242	4.716327	0.029885	1.677649	
Constant	0.231324	0.242587	0.909299	0.340308	1.260268	Chi^2^(1) = 1.7774 *p* = 0.18248
Physical activity (X_5_)	0.291956	0.221111	1.743476	0.186709	1.339044	
Constant	0.068993	0.214549	0.103409	0.747779	1.071429	Chi^2^(1) = 9.8314 *p* = 0.00172 *
Gender (X_6_)	1.23934	0.41601	8.875087	0.002893	3.453333	
Constant	−1.68933	0.635784	7.06004	0.007886	0.184644	Chi^2^(1) = 13.983 *p* = 0.00018 *
Age (X_7_)	0.046534	0.013659	11.6069	0.000658	1.047633	
Constant	−0.20764	0.373399	0.309224	0.578161	0.8125	Chi^2^(1)=4.0865 *p* = 0.04323 *
Place of residence (X_8_)	0.858227	0.426263	4.053674	0.044084	2.358974	
Constant	0.163114	0.222583	0.537033	0.46367	1.177171	Chi^2^(1) = 4.5347 *p* = 0.03322 *
Education (X_9_)	0.504959	0.244828	4.25392	0.039168	1.656918	
Constant	0.049956	0.376678	0.017589	0.894492	1.051225	Chi^2^(1) = 1.4747 *p* = 0.22462
Professional work (X_10_)	0.382224	0.317063	1.453266	0.228014	1.46554	
Constant	0.275687	0.275348	1.002464	0.316723	1.317436	Chi^2^(1) = 0.71413 *p* = 0.39808
Stress (X_11_)	0.186499	0.221261	0.710463	0.399297	1.205023	
Constant	1.283882	0.50388	6.492258	0.010839	3.610627	Chi^2^(1) = 3.3167 *p* = 0.06859
Inventory of Health Behaviours (IHB) (X_12_)	−0.57203	0.319851	3.198447	0.073718	0.56438	
Constant	1.093376	1.189821	0.844454	0.358133	2.984332	Chi^2^(1) = 0.29765 *p* = 0.58536
Psychosocial factors (X_13_)	−0.03365	0.061944	0.295053	0.587004	0.966912	

Note. * Chi-squared test; *p*—statistical value; * *p* ≤ 0.05. The remaining predictors were grouped into categories and coded: (1) Lifestyle: X_2_—smoking (0—no, 1—yes), X_3_—drinking alcohol (0—not drinking, 1—occasionally, 2—frequently), X_4_—BMI (0—normal body weight, 1—overweight, 2—obesity), X_5_—physical activity (0—high, 1—medium, 2—low); (2) Sociodemographic variables: X_6_—sex (0—woman, 1—man), X_7_—age (quantitative variable), X_8_—place of residence (0—countryside, 1—city), X_9_—education (0—higher, 1—secondary, 2—vocational and lower), X_10_—professional work (0—physical, 1—mental, 2—unemployed); (3) Other variables: X_11_—level of stress (0—low, 1—medium, 2—high), X_12_—IZZ (0—low, 1—medium, 2—high), X_13_—psychosocial factors (quantitative variable).

**Table 12 ijerph-18-00309-t012:** Logistic regression score (LRS)—assessment of the relationship between selected food products considered individually and the probability of an increased level of homocysteine for the CA group.

Parameter	Estimate	Standard Error	Chi-Square (Walda)	*p*	Odds Ratio (OR)	Statistics for the Model
Constant	0.548383	0.534244	1.053632	0.30468	1.730453	Chi^2^(1) = 0.03416 *p* = 0.85336
Legumes (X_14_)	−0.07146	0.387114	0.034075	0.853549	0.931035	
Constant	−0.60572	0.543432	1.242373	0.265023	0.545682	Chi^2^(1) = 4.3289 *p* = 0.03748 *
Brown bread (X_15_)	0.80866	0.398843	4.110809	0.042618	2.244898	
Constant	1.187843	0.594322	3.994617	0.045654	3.28	Chi^2^(1) = 1.7183 *p* = 0.18992
Fat milk (X_16_)	−0.47	0.360601	1.698821	0.192452	0.625	
Constant	−0.52057	0.552626	0.887338	0.346207	0.594184	Chi^2^(1) = 3.4567 *p* = 0.06301
Multi-grain flakes (X_17_)	0.665748	0.361292	3.395507	0.065384	1.945946	
Constant	−0.9018	0.55569	2.633636	0.104632	0.405838	Chi^2^(1) = 6.6469 *p* = 0.00994 *
Wheat pasta, groats (X_18_)	0.930788	0.367197	6.425452	0.011254	2.536508	

Note. * Chi-squared test; *p*—statistical value; * *p* ≤ 0.05. Category: Food products X_14_—X_18_ was coded (0—yes, 1—no).

**Table 13 ijerph-18-00309-t013:** Logistic regression score (LRS)—assessment of the relationship between variables considered individually and the probability of an increased homocysteine level for the SDA group.

Parameter	Estimate	Standard Error	Chi-Square (Walda)	*p*	Odds Ratio (OR)	Statistics for the Model
Constant	cannot be estimated					
Smoking cigarettes (X_2_)						
Constant	cannot be estimated					
Drinking alcohol (X_3_)						
Constant	−0.25049	0.244163	1.05252	0.304935	0.778417	Chi^2^(1) = 0.08538 *p* = 0.77013
BMI (X_4_)	0.07688	0.263015	0.085441	0.770056	1.079913	
Constant	−0.10969	0.229939	0.22757	0.633335	0.896111	Chi^2^(1) = 0.47512 *p* = 0.49065
Physical activity (X_5_)	−0.19082	0.278234	0.470375	0.49282	0.826278	
Constant	−0.187212	0.231963	0.65137	0.419629	0.829256	Chi^2^(1) = 8.1242 *p* = 0.00437 *
Gender (X_6_)	1.13629	0.411589	7.62171	0.00577	3.115265	
Constant	1.117417	0.682945	2.677072	0.101813	3.056949	Chi^2^(1) = 4.2171 *p* = 0.04003 *
Age (X_7_)	−0.02495	0.012492	3.988586	0.045818	0.975361	
Constant	0.154151	0.321196	0.230331	0.631282	1.166667	Chi^2^(1) = 1.8740 *p = 0*.17103
Place of residence (X_8_)	−0.53856	0.394587	1.862886	0.172302	0.583587	
Constant	−0.22749	0.293536	0.600645	0.438338	0.796527	Chi^2^(1) = 0.01059 *p* = 0.91803
Education (X_9_)	0.02358	0.229353	0.01057	0.918114	1.02386	
Constant	0.416838	0.297579	1.962137	0.161295	1.517156	Chi^2^(1) = 7.4577 *p* = 0.00632 *
Professional work (X_10_)	−0.65165	0.2451	7.068639	0.007848	0.521188	
Constant	−0.314	0.282131	1.23865	0.26574	0.730522	Chi^2^(1) = 0.26917 *p* = 0.60389
Stress (X_11_)	0.129151	0.249066	0.268885	0.604084	1.137862	
Constant	−0.80939	0.832888	0.944363	0.331166	0.445131	Chi^2^(1) = 0.56631 *p* = 0.45174
Inventory of Health Behaviours (IHB) (X_12_)	0.339384	0.454064	0.558661	0.454806	1.404082	
Constant	0.4039324	−0.03069895	0.4039324	−0.03069895	0.4039324	Chi^2^(1) = 0.15724 *p* = 0.69172
Psychosocial factors (X_13_)	1.544653	0.0774489	1.544653	0.0774489	1.544653	

Note. * Chi-squared test; *p*—statistical value; * *p* ≤ 0.05. The remaining predictors were grouped into categories and coded: (1) Lifestyle: X_2_—smoking (0—no, 1—yes), X_3_—drinking alcohol (0—not drinking, 1—occasionally, 2—frequently), X_4_—BMI (0—normal body weight, 1—overweight, 2—obesity), X_5_—physical activity (0—high, 1—medium, 2—low); (2) Sociodemographic variables: X_6_—sex (0—woman, 1—man), X_7_—age (quantitative variable), X_8_—place of residence (0—countryside, 1—city), X_9_—education (0—higher, 1—secondary, 2—vocational and lower), X_10_—professional work (0—physical, 1—mental, 2—unemployed); (3) Other variables: X_11_—level of stress (0—low, 1—medium, 2—high), X_12_—IZZ (0—low, 1—medium, 2—high), X_13_—psychosocial factors (quantitative variable).

**Table 14 ijerph-18-00309-t014:** Logistic regression results (LRS)—assessment of the relationship between selected food products considered individually and the probability of an increased level of homocysteine for the SDA group.

Parameter	Estimate	Standard Error	Chi-Square (Walda)	*p*	Odds Ratio (OR)	Statistics for the Model
Constant	−1.94591	0.7559674	6.625817	0.010055	0.142857	Chi^2^(1) = 8.9016 *p* = 0.00285 *
Legumes (X_14_)	1.94591	0.781624	6.197982	0.012795	7	
Constant	0.941669	0.700373	1.80775	0.178788	2.564258	Chi^2^(1) = 3.1204 *p* = 0.07733
Brown bread (X_15_)	−1.02014	0.610478	2.792416	0.094721	0.360544	
Constant	−0.73603	0.878969	0.701203	0.402386	0.479012	Chi^2^(1) = 0.38949 *p* = 0.53257
Fat milk (X_16_)	0.294197	0.474235	0.384848	0.535024	1.342048	
Constant	−0.91191	0.559816	2.65344	0.103336	0.401758	Chi^2^(1) = 1.8149 *p* = 0.17793
Multi-grain flakes (X_17_)	0.537212	0.399963	1.804064	0.179231	1.71123	
Constant	0.103819	0.582921	0.03172	0.858644	1.1094	Chi^2^(1) = 0.31170 *p* = 0.57664
Wheat pasta, groats (X_18_)	−0.25464	0.458224	0.308819	0.578409	0.775194	

Note. * Chi-squared test; *p*—statistical value; * *p* ≤ 0.05. Category: Food products X_14_—X_18_ was coded (0—yes, 1—no).

## Data Availability

The data presented in this study are available on request from the corresponding author. The data are not publicly available due to ethical and privacy issues.

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
