# Peer review of "Comparison of Lifestyle of Catholics and Seventh-Day Adventists and the Relationship with Homocysteine as Risk Factor for Cardiovascular Diseases, a Cross-Sectional Study in Polish Males and Females"

_ijerph, 2021, doi:10.3390/ijerph18010309_

Round 1

Reviewer 1 Report

The authors in the present manuscript compare the lifestyle of two religious groups and try to draw a relationship between lifestyle and level of homocysteine which is an established risk factor for CVD.

Major Comments:

1. In the methods, section authors leave out important questions in terms of alcohol consumption such as for example if the Participants were asked about the types of alcohol they consumed in a week, the frequency of consumption in a week, and the number of servings on a typical day.

2. Straightaway in section 3.3. Addictions the authors classify the participants under no consumption, very rare consumption, rare, frequent and every day consumption. But they fail to mention how this classification was achieved.

3. The authors also do not mention the inclusion/exclusion of any medical history of cardiovascular disease, cerebrovascular disease, and pulmonary disease.  In case cohorts with medical history were included will have an impact on the homocysteine levels irrespective of their religious faith and this shall introduce a strong bias in the study, given the low number of sample size.

4. Another major shortcoming of the study is the lack of suitable statistical methodology. The authors should use logistic regression to conduct the cross-sectional analyses adjusted for age, gender, occupation, marital status, BMI, relevant health habits, and clinical factors, etc. Religiosity should be analyzed as a categorical variable.

5. Adjusted odds ratios for binominal outcomes of cardiovascular risk factors should be calculated.

6. Thorough checkup of typing mistakes (such as line 143 p=0.05) and correction of English language required (Lines 4-6, Lines 202-203)

Author Response

Dear Reviewer,

thank you very much for your comments,

Authors

Reviewer 2 Report

In this manuscript titled, "The Comparison of Lifestyle Seventh-Day Adventist and Catholics Living in Southern Poland. Relationship Lifestyle with the Level of Homocysteine and Risk Factors for Cardiovascular Diseases.", Anna Majda et al., authors aimed to compare the lifestyle of Seventh-day Adventists (SDA) and Catholics (CA), to determine the relationship between the lifestyle of SDA, CA and the level of serum homocysteine as a risk factor for cardiovascular diseases (CVD). Overall, the manuscript is written clearly.  However, the manuscript appears preliminary.

  1. In the study group, SDA ages are 24 to 94 years and CA ages are 20-96 years. As we all know, young persons and senior persons have different lifestyles, such as young persons maybe study at school or early stage of their career, they are more activity while senior persons are retired less activity. Is there any difference in homocysteine levels between young and senior persons in the same group?
  2. Authors need to supply the detailed methods that they used to do analysis in this study.
  1. The manuscript contains several typos that should be corrected. Line 143, ‘p = 0,05’ should be ‘P < 0.05’; All of p-vale should be Capital letter P with italic.

Author Response

Dear Reviewer,

thank you very much for your comments,

Autors

Reviewer 3 Report

This manuscript compares two religious groups (catholics and eventh day adventists) and their lifestyles ( physical activity, nutrition, smoking,stress, homocysteine and personality).The number of participants are relatively small in comparison with the big heterogenity of their age (between 18 and 53  years).

Title is not clear, suggestion: Comparison of lifestyle of Catholics and Seventh Day Adventists and the relationship with homocysteine as a riskfactor for Cardio Vascular Diseases, a crosssectional study in Polish males and females

The abstract is not clear about author's questionnare (?).  What is meant with äddiction". What is :results deviated from the norms? I expect to see significant differences between lifestyles.

Introduction is oke and explains the the two purposes: comparison of the two groups with respect to lifestyles and relationship of homocysteine levels and lifestyles.

In matreial, methods and results there is too much text without tables: I miss table about characteritics of the CA and SDA groups. In results  I miss a table about addiction,physical activity, while the table 1 of nutrition is too detailed (is frequency enough and not how often?). Also not clear what is tested: differences between groups or relation with homocysteine levels.

Moreover if the differences in food are not significant, you can also not expect differences in homcysteine.

Conclusions are vague.

Author Response

(The authors gave the same response as above.)

Round 2

Reviewer 1 Report

I am satisfied with the changes made in the manuscript.

Author Response

Thank you for this rating. Changes were made in the English language, marked in the text of the article.

Line 3 exchange … as a Risk…

….as Risk…

Line 29 exchange ….in CA were significantly different from SDA.

…for CA were significantly different in comparison to SDA.

Line 31 exchange…alcohol consumption…

…drinking alcohol..

Line 33 exchange…in the studied group of CA….

…for the studied group of CA….

Line 36 exchange… The level of homocysteine in the studied group of SDA was significantly influenced by socio-demographic factors: gender, age, professional activity and consumption of legumes.

…For the studied group of SDA, the level of homocysteine was significantly influenced by socio-demographic factors such as gender, age, professional activity, and consumption of legumes.

Lines 117-121exchange In Part A, the questions concerned drinking alcohol (type: wine, beer, vodka, amount in standard portions, frequency: I do not consume it, sporadically/very rare – not more than once a month, rare – several times a month or once a week, often/frequent – several times a week, every day), smoking cigarettes (how many, how many years).

Part A contained questions related to the consumption of alcohol (type: wine, beer, vodka, the number of standard serving sizes consumed and the frequency: I do not drink alcohol, I drink occasionally/very rarely – not more than once a month, rarely – several times a month or once a week, often/frequently – several times a week, daily smoking cigarettes (how many cigarettes daily, for how many years).

Line 240 exchange …declaring consumption of legumes…

…who declared the consumption of legumes…

Line 244 exchange …declaring consumption of full fat…

… who declared the consumption of full fat…

Lines 494-503 exchange

Thanks to the logistic regression model, it was possible to prove in the presented study that in older CA, living in the city, with lower education, and men, the probability of having elevated levels of homocysteine was significantly higher, as expected. On the other hand, in SDA, the risk of elevated homocysteine value decreased in the elderly and the unemployed which was an unexpected result.

Public health professionals and nurses should develop culturally and religiously specific educational interventions that promote healthy lifestyle principles in the prevention of CVD, especially among Catholics.

Thanks to the use of the logistic regression model in the presented study, it was possible to prove that for older CA, living in the city, with lower education, males, the probability of having increased levels of homocysteine as expected was significantly higher. On the other hand, for SDA, the risk of having increased homocysteine value was lower for the elderly, and the unemployed which was an unexpected.

Both public health professionals and nurses should therefore develop educational interventions related to culture and religion in order to promote healthy lifestyle principles in the prevention of CVD, especially among Catholics.

Reviewer 3 Report

The authors made all the changes I have suggested and increased the quality of the  resubmitted paper.

The authors have still to check their text and tables for uniformity in explaining the used abbreviations e.g. BMI in abstract and CA and SDA and others all over in the tables and in description of the tables!

Author Response

Thank you for this valuable comment and rating. Correction made. The authors still have to check the text and tables for uniformity in explaining the abbreviations used

Line 28 in abstract  added

Body Mass Index (BMI)

Line 73 conversion

from ADS to SDA

Line 84 conversion

from ADS to SDA

Line 205 conversion

from Catholics  to CA

Line 248 conversion

from Catholics to CA

Line 251 conversion

from Catholics  to CA

Line 264 conversion

from Catholics  to CA

Line 288 conversion

from SAD to SDA

Line 351 conversion

of Catholics to CA group

Line 368 conversion

of Catholics to CA group

Line 458 conversion

from ADS to SDA

Line 487 conversion

from ADS to SDA

Line 489 conversion

from ADS to SDA

Line 494 conversion

from ADS to SDA

Line 614

Deletion of a number 34

Line 615

Deletion of a number 35